# The Effect of Microwave Radiation on the Green Color Loss of Green Tea Powder

**DOI:** 10.3390/foods11162540

**Published:** 2022-08-22

**Authors:** Huijuan Wang, Yan Zhu, Dongchao Xie, Haihua Zhang, Yahui Zhang, Peng Jin, Qizhen Du

**Affiliations:** Department of Food Science and Technology, College of Food and Health, Zhejiang A & F University, Hangzhou 311300, China

**Keywords:** color, green tea powder, kinetics, microwave radiation

## Abstract

Microwave radiation is one of the main heating methods for food processing, especially affecting the color quality of colorful foods. This work presents the effect of microwave radiation on the green color loss of green tea powder (GTP) by the color description (L*, a*, b*, and H_a_ of green tea powder, L*:whiteness/darkness, a*: redness/greenness, and b*: yellowness/blueness; H_a_ derived from Hunter a and b could visually describe the color space) of the Hunter color system. First, the L*, a*, and b* were determined from the GTP samples treated with various microwave powers with the change of time to investigate the kinetic of color loss. Then, the L*, a*, and b*and temperature of GTP samples with serious thickness treated with constant microwave power (700 W) for a different time were determined to study the effect of sample thickness on the color loss. Finally, the chemicals that contributed to color change in the GTP samples treated with mild, moderate, and severe radiation were analyzed. The results showed that L*, |a*| (|a*|was the absolute value of a*), b*, and H_a_ decreased with the power increase in microwave radiation, and their changes conformed to the first-order kinetics. The activation energies (E_a_) of different thickness GTP for change of L*, a*, b*, and H_a_ values could be predicted with the fitting models, and E_a_ for 20 mm-thick GTP were approximately 1/5, 1/8, 1/8, and 1/13 of those for 4 mm-thick GTP. The color loss was mainly caused by the Mg^2+^ loss of chlorophylls and the formation of derivates under mild radiation, the degradation of chlorophylls and the formation of theaflavin from catechins under moderate radiation, and the degradation of chlorophylls and their derivates accompanied by Maillard reaction between reducing sugar and amino acids under severe radiation. The results indicate that sample thickness and radiation time are two key parameters to keeping the color of GTP in food processing and microwave pasteurization.

## 1. Introduction

Green tea powder (GTP) is very well-received in the food industry because of its comfortable color, complete bioactive components of tea leaves, and excellent characteristics for food processing [1], such as smoothies and baking [2,3,4,5]. GTP contains higher [1,2,3,4,5,6,7,8,9] contents of phenolic acids, theanine, and chlorophylls than other green tea varieties [6]. Moreover, GTP as a powdered form makes it more accessible to be applied in various foods, such as noodles, biscuits, candy, and pastries [7,8,9].

The green color is one of the most important quality factors of GTP, in which a complex mixture of chlorophylls and their derivates are the main contributors. Chlorophylls are highly susceptible to degradation during heat processing or sterilization, resulting in color changes in foods [10]. Especially with the increasing use of GTP in food, the considerable losses in the organoleptic quality of food caused by heating have become a bottleneck for its applications. As well known, microwave radiation is extensively used for drying, heating, and sterilization in food processing [11,12]. The application of GTP for baking foods and the sterilization processing in Matcha (a green tea powder) production is expected to apply microwave radiation. However, few literature reports on the effect of microwave radiation on the chemicals that influence the color of GTP and the kinetics of color loss, which could help control the quality of products.

Hunter color system is extensively used to investigate the color change of foods; it mainly measures three parameters, namely, L* (whiteness/darkness), a* (redness/greenness), and b* (yellowness/blueness), as a versatile and reliable method to assess the color deterioration and quality control in foods [13,14,15]. The hue angle (H_a_) derived from Hunter a and b could visually describe the color space. When performing first quadrant calculations, the result of the H_a_ derivation is highly reliable and easily managed in statistical analysis. By accounting for the full 360° H_a_, virtually any change in hue is easily interpreted and analyzable statistically [16]. The present study aimed to employ the Hunter L*, a*, and b* systems and H_a_ to investigate the dynamic effect of microwave radiation on the change in color and the chemical components of GTP.

## 2. Materials and Methods

### 2.1. Chemicals and Reagents

Green tea powder (GTP) was provided by Shaoxing Royal Tea Village Co. Ltd. (Shaoxing, China). Standards for (−)-epicatechin (EC), (−)-epigallocatechin (EGC), (−)-epicatechin gallate (ECG), (−)-epigallocatechin gallate (EGCG), gallic acid (GA), and theaflavin (TF) were purchased from Sigma−Aldrich (St Louis, MO, USA). Standard chlorophyll-a and -b (Chl-a and Chl-b), pheophytin-a and -b (PP-a and PP-b), pyropheophorbide-a (PyPB-a), and methyl pheophorbide-a were purchased from Shanghai ZZBIO Co., Ltd., China. HPLC-grade ethanol, methanol, and acetonitrile (Merck, Darmstadt, Germany), analytical-grade acetone, and other reagents were purchased from Sinopharm Chemical Reagent Co., Ltd. (Shanghai, China).

### 2.2. Microwave Treatment of Green Tea Powder

Microwave treatment of GTP was performed by two microwave ovens with technical features of 220 V, 50 Hz, and a frequency of 2450 MHz. The Midea L213B microwave oven (Midea Group, Fushan, China) could output a constant power of 700 W, and the C-Wiseoven 30PX98 microwave oven (C-Wiseoven Co., Ltd., Zhongshan, China) could output five powers (200, 400, 600, 800, and 1000 W). During microwave radiation experiments, each sample contained in a Petri dish with an 11.5 mm inner diameter was placed in the center of the rotating glass plate. GTP samples (2 mm thickness) were treated with various microwave power (200, 400, 600, 800, and 1000 W) with the change of time to investigate the kinetic of color loss. GTP samples with a serious thickness (4, 8, 12, 16, and 20 mm) were treated with a constant microwave power (700 W) at different times to study the effect of sample thickness on color loss. The sample thickness of 4, 8, 12, 16, and 20 mm GTP correspond to the sample amount of 20.8, 41.6, 62.4, 83.2, and 104.0 g, respectively. The initial sample at room temperature (22–25 °C) with previous microwave treatment in each section of the experiments was regarded as a control (CK).

To analyze the chemicals that contributed to color change, GTP samples (2 mm thickness) were treated with mild (4 min), moderate (6 and 9 min), and severe radiation (12 and 15 min) with microwave (1000 W) to yield samples with various a* values (−16.72, −12.17, −8.66, −6.01, −0.23) and various b* values (39.56, 37.24, 36.05, 34.78, 32.97).

Three replications of each experiment were performed, and the reproducibility of the assayed values was within the range of 2.5%.

### 2.3. Temperature Determination of Green Tea Powder in Microwave Oven

A large sample thickness causes the high temperature in the microwave radiation experiment of GTP. To study the apparent activation energy required for color change in different thicknesses of green tea powder, a fiber optic temperature sensor FOT-L-SD-R1 (FISO Technologies Inc., Quebec City, QC, Canada) was employed to detect the temperature of GTP after microwave radiation for the samples with a serious thickness (4, 8, 12, 16 and 20 mm) treated with various microwave power (200, 400, 600, 800, and 1000 W) for a given time. The sensor was placed into the GTP at the center of the sample to show the temperature after the given time of radiation. Three replications of each experiment were performed.

### 2.4. Color Analysis

The CIE L*, a*, and b* values of GTP samples were measured by a spectrophotometer (ColorQuest XE, HunterLab, Reston, VA, USA). The instrument was calibrated with a white ceramic plate (X = 93.50, Y = 0.3114, Z = 0.3190) and then used for the scanning of L*, a*, and b* of the treated GTP samples. Five times scanning was performed per sample. The average value was recorded as L*, a*, and b* values. The measured L*, a*, and b* values were calibrated by coefficients because the calculated H_a_ from the a* and b* values did not match the color of the GTP. For example, the H_a_ value of fresh GTP was only 104.0^°^ (yellow, Appendix A), which was clearly different from that of yellow-green GTP.

For calibration of the measured values of L*, a*, and b*, a series of GTP samples was radiated by microwave to yield samples with different values of L*, a*, and b*. The GTP samples were transformed into a physical image with a Canon 200D camera (Canon (China) Co. Ltd. Beijing, China). Then, the color was matched with a*, LScolor software V10.14 (Shenzhen Linshang Technology Co., Ltd., Shenzhen, China), which showed matched values of Hunter L*, a*, and b*. The calibrated values of L*, a*, and b* were defined as L*, a*, and b*, respectively, which were calculated by their calibration coefficients (Appendix A).

H_a_ was calculated as follows:H_a_ = arctan (b*/a*)(1)

### 2.5. Kinetic Analysis of Color Loss by Microwave Radiation

The L*, a*, b*, and H_a_ obtained from the determination of the GTP samples treated with a constant microwave power (700 W) with the change of time were fitted with an exponential model (2) to decide the kinetic of color loss.
Y = Y_0_ + A exp(−kt)(2)
where Y is the values of the L*, a*, b*, and H_a_, respectively, A is a frequency factor, k is the rate coefficient of the color value change, and t is the processing time of microwave radiation.

The color kinetic change (L*, a*, b*, or H_a_) depending on temperature due to the change of sample thickness was fitted by the Arrhenius Equation (3) [17].
k = k_0_ exp(−E_a_/RT)(3)
where k is the rate coefficients of the color value change at a corresponding temperature (T); k_0_ is the frequency factor (min^−1^); E_a_ is the activation energy for color loss (kJ/mol); R is the universal gas constant (8.314 KJ/mol), and T is the absolute temperature (K). The k values were obtained by Equation (4) as follows:Ln(Y/Y_0_) = −kt (4)
where Y_0_ is the initial value of L*, a*, b*, or H_a_, and Y is the L*, a*, b*, or H_a_ value of GTP radiated for t minutes. Therefore, E_a_/R can be obtained by fitting the k values of the L*, a*, b*, or H_a_ and 1/T with the exponential Equation (3).

### 2.6. Analytical Methods

Chlorophylls: The chlorophylls of the GTP samples were determined with the colorimetric method as described by Huang et al. (2007), with slight modification to analyze the correlation between the L*, a*, b*, or H_a_ value and chlorophylls [18]. In Brief, GTP (0.2 g) was mixed with 25 mL of 80% (*v*/*v*) acetone for 5 min on a tube mixer and then filtered through Whatman No. 1 filter paper (GE Healthcare Rydalmere, NSW, Australia). The amounts of chlorophyll were calculated in accordance with the absorption at 663 and 645 nm by using the following formulas:Chlorophyll a (mg/L) = 12.7A_663_ − 2.95A_645_(5)
Chlorophyll b (mg/L) = 22.9A_645_ − 4.67A_663_(6)
Total chlorophyll content (mg/L) = Chlorophyll a + Chlorophyll b(7)

For the investigation of the change in chlorophylls by microwave radiation, the chlorophylls and their deviates in the treated GTP samples with various degrees of color loss were determined by high performance liquid chromatography (HPLC) (Waters E2695 system and Waters 2998 DAD detector, Waters Corporation, Milford, MA, USA) and a SinoChrom ODS-BP column (4.6 mm I.D. × 25 cm, Elite Analytical Instrument Co., Ltd., Dalian, China).

For the extraction of chlorophylls in GTP, a 100 mg sample in a 50-mL centrifuge tube was added with 15 mL of 85% acetone. The mixture was homogenized (Kinematica Co, Ltd., Malters, Switzerland) for 5 min and then centrifuged for 5 min at 3000 rpm. The supernatant solution was transferred to a 50-mL volumetric flask. The above extraction was repeated twice on the residue. The extract was made to the mark with 85% acetone and then filtered through a 0.5 µm membrane filter (Waters (Shanghai)Technology Co., Ltd., Shanghai, China) for HPLC injection. The chlorophylls of GTP were separated by the mobile phase composed of solvent A (95% ethanol (*v*/*v*) containing 0.005 M sodium chloride) and solvent B (80% ethanol (*v*/*v*) containing 0.005 M sodium chloride) at a linear rate ranging from 5:95 mixture to 95:5 mixture over 10 min. The 95:5 mixture was run isostatically for an additional 25 min. The flow rate was 1.0 mL/min, and it was monitored at 425 nm [19].

Catechins & theaflavins: One g sample was extracted two times with 20 mL of 70% methanol (*v*/*v*) at 70 °C for 10 min. The methanol extract solutions were combined into a 50 mL volumetric flask and adjusted to scale with cold 70% (*v*/*v*) methanol for HPLC injection with the above HPLC system. For the analysis of catechins, the mobile phase consisted of a solution of A (0.05% (*v*/*v*) trifluoroacetic acid in water) and B (99.9% acetonitrile) run with 87% solution A and 13% solution B from 0 min to 25 min at a flow rate of 1.0 mL/min and monitored at 254 nm [20]. For the analysis of theaflavins, the mobile phase consisted of solvent A (9% acetonitrile, 2% acetic acid, and 89% water, *v*/*v*) and solvent B (80% acetonitrile, 2% acetic acid and 28% water, *v*/*v*). Elution was performed with 100% A from 0 to 10 min, 100% A to 68% A and 0% B to 32% B from 10.01 to 25 min by linear mode, 68% A and 32% B from 25 min to 35 min, and 68% A and 32% B to 100% A and 0% B by linear mode from 35 to 40 min. The detected wavelength was 278 nm [21].

Reducing sugars & total free amino acids: The content of reducing sugars in GTP samples was quantified with the 3,5-dinitrosalicylic acid colorimetric, spectrophotometric method [22] with modification. Prior to the extraction of reducing sugars, the polyphenols and chlorophylls in GTP samples (0.5 g) were thoroughly removed by extracting five times at 60 °C for 20 min with 25 mL ethanol–acetone (3:7, *v*/*v*). The residue was subjected to extraction of reducing sugars with hot water (60 °C, 30 min). The total amount of free amino acids in GTP was determined by the ninhydrin method [23].

### 2.7. Statistical Analyses

OriginLab 2019 software (OriginLab Corporation, Northampton, MS, USA) was used for the fitting analysis of the measured values to the hypothetical models. Microsoft Excel was used to deal with data. Bivariate correlations between chlorophylls and color values were performed using SPSS PASW Statistics software (version 18; IBM, Armonk, New York, NY, USA).

## 3. Results and Discussions

### 3.1. Kinetic of Color Loss in Green Tea Powder under Different Microwave Powers

To investigate the effect of microwave output power on the color change kinetics of GTP, GTP was radiated with five microwave output powers at different times. As shown in Figure 1, the brightness of GTP (L*) decreased with the increase in microwave power from an initial value of 55.85 ± 0.80. When radiation time reached 30 min, the L* of the GTP radiated under 200 W only decreased to 55.30 ± 0.63, and the GTP showed no color change in the naked eye. However, the L* of the GTP radiated under high power speeded up the lowering of L*. When L* decreased to approximately 52, the GTP was obviously darker than the original (Figure 1A). The change in L* with radiation time under all the powers conformed to the first-order kinetic with a coefficient of determination (R^2^) of more than 0.98 (Table 1). The initial a* and b* of GTP without radiation treatment (CK) were 20.00 ± 0.46 and 39.70 ± 0.74 (Figure 1A,B), respectively, which gave a H_a_ of 116.47. The L* plus H_a_ yielded a GTP color consistent with the color at the color spectrum (Appendix A). The value of H_a_, which showed no color change in the naked eye, was lower than 115° compared with the values in CK (Figure 1D). Similar to the change in L*, the change in a*, b*, and H_a_ with radiation time under all the powers conformed to the first-order kinetic with R^2^ more than 0.98 and lower standard error (σ) of kinetic rate constant (k, Table 1), except that radiated with 200 W. The kinetic change in L*, a*, b*, and H_a_ agreed with those in tea leaves dried by microwave drying [1]. The results obtained in the present study were in agreement with those of studies published in the literature, and several authors have stated that a first-order kinetic model better fitted the L*, a*, b*, values of tomato puree [24], basil [25], celery leaves [26], and spinach [27]. Our results suggest that the color loss can be predicted if the GTP is treated with microwave, such as for microwave sterilization, which can help us to control the product quality.

### 3.2. Low Apparent Activation Energy Required for Color Loss of Thick GTP Sample

The thickness of the processed sample by microwave radiation is one of the most important influencing factors of the heating effect. In the present study, the L*, a*, and b* values of the GTP with thicknesses (T_Th_) of 4, 8, 12, 16, and 20 mm were obtained, and the final temperature in the interior of the GTP was measured with 700 W of microwave power. As shown in Table 2, the GTP temperature T_m_ increased rapidly with the increase in thickness. The final temperature of 4-, 8-, and 12 mm-thick GTP radiated for 380, 330, and 280 s was less than 100 °C, as the corresponding GTP samples received radiation energy of 3.55, 1.54, and 0.87 W/g, respectively. Meanwhile, the temperature of 16- and 20-mm thick GTP radiated for 230 and 180 s reached 119.5 ± 4.16 °C and 132.8 ± 4.73 °C, while the received radiation energy of the corresponding GTP samples were only 0.54 and 0.28 W/g, respectively. Correspondingly, the values of L*, |a*|, and b* increased significantly with an increase in GTP thickness. The color of the GTP sample with thicknesses of 12, 16, and 20 mm radiated for 280, 90, and 150 s obviously changed from green to yellow at corresponding temperatures of 96.7 ± 3.56 °C, 101.2 ± 3.08 °C, and 113.6 ± 5.62 °C, respectively. Therefore, GTP must be made under less than 90 °C during food processing, and a low GTP thickness in microwave pasteurization must be selected to maintain its green color. The GTP temperature only increased to 31.5 ± 1.37 °C though the sample received radiation of 0.93 W/g, which was only slightly higher than the CK sample temperature (25 °C). Correspondingly, the color showed almost no change.

Up to date, GTP has been added to noodles, biscuits, candy, and pastries. GTP noodle maintained its light-green color because the noodle only needed 35–45 °C for 90 min to dry [28]. On the contrary, GTP roulade cake and GTP cake did not show satisfactory green color due to the required temperatures of 160 °C for 33 min or 165 °C for 18 min [29,30,31].

The activation energy for the color change of GTP is defined as apparent activation energy (E_a_) because GTP has no molecular weight, such as chemical molecules. The E_a_ required for change in L*, a*, b*, and H_a_ is shown in Table 3. The results indicated that the E_a_ for change in L*, a*, b*, and H_a_ increased with the increase in GTP thickness (H_Th_). The E_a_ values for 20 mm-thick GTP were approximately 1/5, 1/8, 1/8, and 1/13 of those for 4 mm-thick GTP in terms of L*, a*, b*, and H_a_, respectively. This finding revealed that decreasing GTP thickness during food processing and microwave pasteurization could effectively slow the color loss. The E_a_ for change in L*, a*, b* values of GTP with the thickness could be predicted with the fitting models in Table 3.

The sample amount for 4, 8, 12, 16, and 20 mm thick GTP in the present study was 20.8, 41.6, 62.4, 83.2, and 104.0 g, respectively. Evidently, higher GTP temperature yielded lower radiation energy per unit weight when we compare the data of the thicker samples with the thinner samples in Table 2. As we know, the traditional method of heating materials is based on the principles of heat conduction, heat convection, and heat radiation. The heat is gradually transferred from the material surface to the interior to heat it. By contrast, microwave heating converts microwave energy into heat energy through the dissipation of the motion, resulting in higher temperature in the interior than in the surface layer of the materials due to the strong penetration ability and the weak dissipation in the interior [32]. The thick sample made the heat in the interior difficult to escape. The thick sample made the heat in the interior difficult to escape, resulting in higher temperature values of the thick samples than those of the thin samples. Consequently, the higher temperature in the interior of the thicker GTP sample was easy to cause the chemistry change impacting the color of GTP. Therefore, the activation energy for the color change of the thick GTP sample was lower than that of the thin GTP sample.

### 3.3. Chemicals to Contribute Color Loss in Green Tea Powder under Microwave Radiation

The contents of chlorophyll a (Chl-a) and chlorophyll b (Chl-b) in all the samples treated by microwave radiation was measured to understand the effect of chlorophylls on the color change in GTP under microwave radiation. Figure 2 shows the regression analysis between hunter L*, a*, and b* and H_a_ with Chl-a, Chl-b, or the total of both (Chl-T). The results indicated the correlation coefficients were 0.9878, 0.9834, and 0.9909 between a* and Chl-a, chl-b, and Chl-T, respectively, indicating a decline in a* (redness/greenness) majorly caused by the decrease in chlorophylls. The correlation coefficients were 0.9654, 0.9573, and 0.9668 between H_a_ and Chl-a, chl-b, and Chl-T, respectively, suggesting that H_a_ was majorly influenced by chlorophylls and minorly influenced by the change in other compounds, which could lead to change in b* as H_a_ is derived from a*and b*. The correlation coefficients were 0.8937, 0.8853, and 0.8954 between b* and Chl-a, Chl-b, and Chl-T, respectively, indicating that the effect of chlorophylls on b* was less than that on a*. In addition, b* was possibly influenced by the derivates of chlorophylls or other substances because b* responds to the property of yellowness/blueness. Low correlation coefficients of 0.6637, 0.6749, and 0.6692 were found between L* and Chl-a, chl-b, and Chl-T, respectively, suggesting that the compounds that made GTP dark were yielded in microwave radiation. Accordingly, detecting the derivates of chlorophylls, catechins, free amino acids, and reducing sugars is necessary because oxidation, polymerization, and the Maillard reaction result in olive, yellow, brown, or deep color.

Chlorophylls can usually derivate the isomers and their derivates (Figure 3A). In fresh GTP (CK), Chl-a′ and Chl-b′, which are the isomers of Chl-a and Chl-b, were detected, possibly due to the heating during milling and steaming because GTP was processed from the milling of a steamed green tea. In addition, pheophytin a (PP-a); pheophytin b (PP-b); pheophorbide-a (PB-a); their isomers PP-a′, PP-b′, and PB-a′; methyl pheophorbide a (MePB-a); and pyropheophorbide a (PyPB-a) were detected (Figure 3B). The change in chlorophylls and their derivates in GTP (thickness of 2 mm) under microwave radiation of 1000 W is shown in Figure 3C. The contents of these compounds in the GTP samples with microwave radiation times of 4, 6, 9, 12, and 15 min are given in Figure 3C. Compared with the fresh GTP, Chl-a decreased by 7.9%, 25.4%, 66.2%, 77.5%, and 94.1%, and Chl-b decreased by 7.3%, 23.6%, 39.1%, 64.5%, and 95.5% for those samples with radiation times of 4, 6, 9, 12, and 15 min, respectively. This finding indicated that Chl-b was more stable than Chl-a to some degree, in agreement with the studies on the thermostability of chlorophylls [33,34,35,36]. From 0–12 min, Chl-b′ exhibited an increasing trend with the increase in the time of microwave radiation, implying that a part of Chl-b converted into Chl-b′ possessed high stability. PP-a showed an increasing trend during moderate microwave radiation due to the loss of magnesium of Chl-a by heating. However, overheating degraded PP-a and caused it to decrease during excessive microwave radiation. Furthermore, HPLC results showed that PP-b was converted into PP-b′, similar to Chl-b, but PP-b′ was degraded only under severe microwave radiation. PP-a, PP-a′, PP-b, PP-b′, and MePB-a could fade the green color of GTP because they yield olive-green color. PyPB-a exhibited a high content in fresh GTP, which could contribute to the brownish-greying color of GTP. However, PyPB-a also decreased with time of microwave radiation. Therefore, the decrease in chlorophylls should be the major factor in the color loss of GTP, and the green color of fresh GTP gradually changed to olive green from 0–12 min. In the sample radiated for 15 min, all the chlorophylls and pheophytins decreased, thus inducing the GTP to have a brownish-yellow color (Figure 4).

GTP is rich in catechins, which could be oxidized and polymerized to theaflavins and thearubigins that possibly influence the color of GTP radiated by microwave radiation. Moreover, the Maillard reaction between reducing sugars and free amino acids could form brown color substances. Therefore, the changes in gallic acid, catechins, reducing sugars, and free amino acids were further investigated. As shown in Table 4, EC, EGC, and free amino acids (FAA) showed a significant reduction in the sample radiated for 12 min. Meanwhile, all the chemicals exhibited a significant decline in the sample radiated for 15 min. Theaflavins were detected by HPLC to confirm whether the formation of theaflavins from catechins influences the color of GTP (Appendix A) because theaflavins could be produced by oxidative coupling with radicals as an oxidizing agent [37]. The results show theaflavin from epigallocatechin and epicatechin were detected after microwave radiation for 12 and 15 min, but the content at 15 min was less than that at 12 min, which indicates theaflavin (yellow) contributes to the green color loss of GTP under severe radiation. The decline in FAA and reducing sugars suggested that the Maillard reaction occurred between them, which possibly contributed to the dark color of the GTP because the color formation via the Maillard reaction is attributed mainly to melanoidins [38].

Based on the above results, the color change in GTP under microwave radiation was mainly caused by the Mg^2+^ loss of chlorophylls, the formation of the derivates under mild radiation, the degradation of chlorophylls and their derivates at moderate radiation, and the degradation of chlorophylls and their derivates accompanied by the formation of Maillard products and theaflavin and degradation of theaflavin.

## 4. Conclusions

The green color of green tea powder was gradually lost via microwave radiation with increasing time and thickness of the GTP sample. L*, |a*|, b*, and H_a_ decreased with the power increase in microwave radiation, and the changes conformed to the first-order kinetic. The estimated activation energies for change of the L*, a*, b*, and H_a_ values for 20 mm-thick GTP were approximately 1/5, 1/8, 1/8, and 1/13 of those for 4 mm-thick GTP. The fitted model of H_a_ versus H_Th_ could be used for color prediction in the related cases. The color change was mainly caused by the Mg^2+^ loss of chlorophylls, the formation of the derivates under mild radiation, the degradation of chlorophylls and their derivates at moderate radiation, and the degradation of chlorophylls and their derivates accompanied by the formation of Maillard products and theaflavin and then degradation of theaflavin. These results indicated that food heat processing and microwave pasteurization could cause green color loss.

## Figures and Tables

**Figure 1 foods-11-02540-f001:**
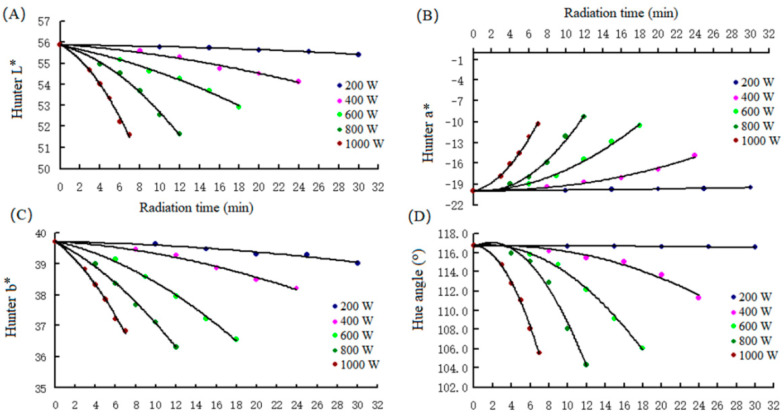
L* (**A**), a* (**B**), b* (**C**), and H_a_ (**D**) values of green tea powder samples at different microwave power with various radiation times. All the determination values from the three experiments gave an RSD of 0.4−2.5%.

**Figure 2 foods-11-02540-f002:**
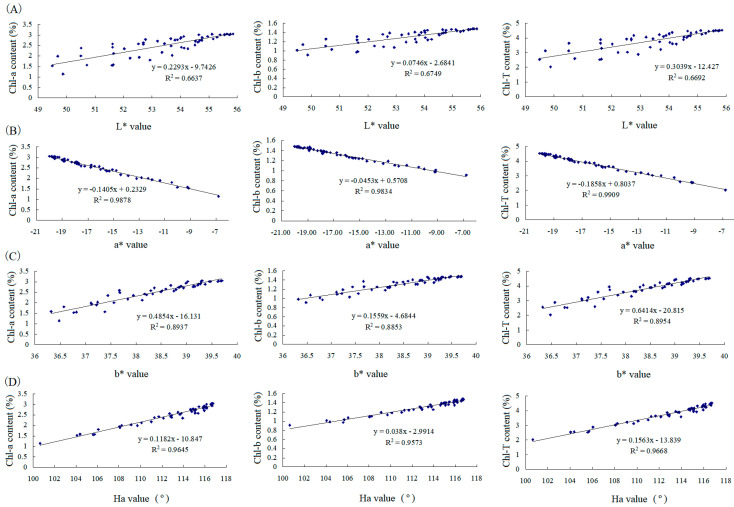
Regression analysis between the L*, a*, b* or H_a_ and chlorophyll a (Chl−a), chlorophyll b (Chl−b), and total chlorophylls (Chl−T). (**A**): L* with Chl-a, Chl-b, and Chl-T; (**B**): a* with Chl-a, Chl-b, and Chl-T; (**C**): b* with Chl-a, Chl-b, and Chl-T; (**D**): H_a_ with Chl-a, Chl-b, and Chl-T.

**Figure 3 foods-11-02540-f003:**
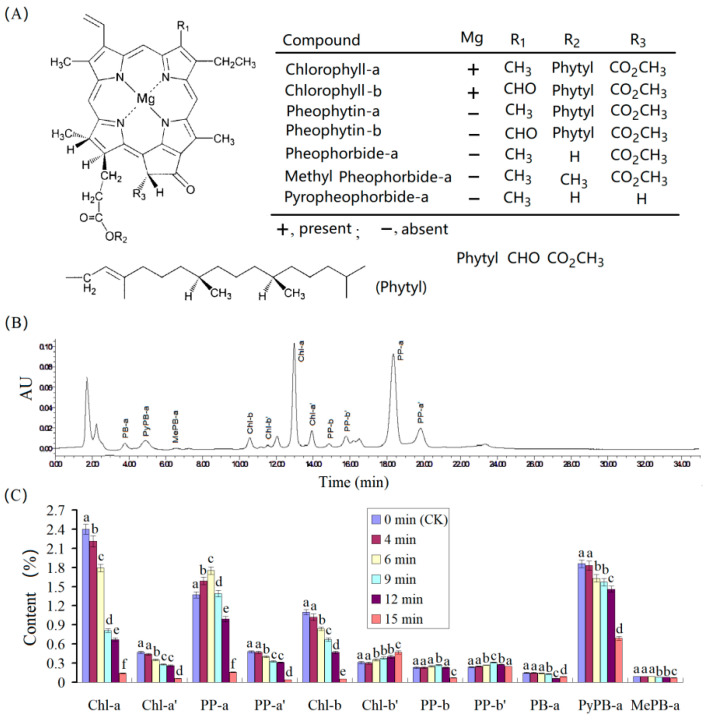
The chlorophylls and their derivates, and their change in green tea powder after microwave radiation of various times. (**A**) structures of chlorophylls and their derivates, (**B**) HPLC chromatogram of chlorophylls and their derivates in fresh green tea powder, and (**C**) change of chlorophylls and their derivates after microwave radiation; the different letters over the bars of a group indicate a significant difference (*p* < 0.05).

**Figure 4 foods-11-02540-f004:**
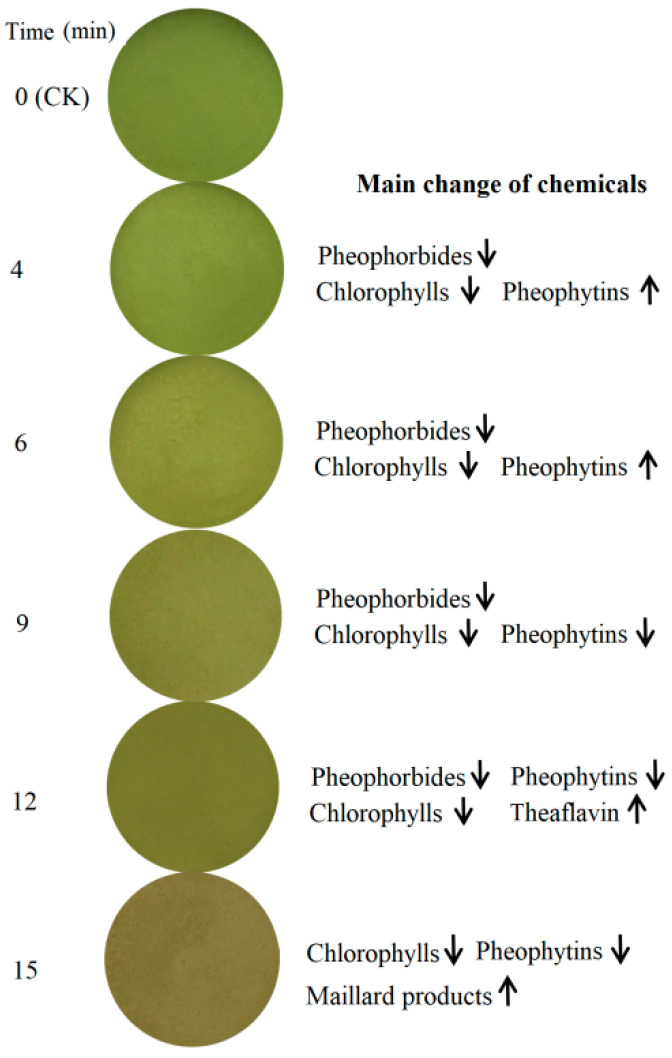
The main change of chemicals contributed to green color loss under green tea powder after microwave radiation of various times.

**Table 1 foods-11-02540-t001:** The estimated kinetic parameters and the statistical values of kinetic models for L*, a*, b*, and Hue angle (H_a_) for various microwave output powers. Y – Y_0_ = A exp(-kt).

Parameter	Power (w)	Y_0_		A		K (m^−1^)		R^2^
		Value	σ	Value	σ	Value	σ	
L*	200	55.94	0.040	−0.096	0.0297	0.0578	0.00836	0.9964
	400	57.02	0.180	−1.128	0.1635	0.0399	0.02062	0.9835
	600	57.83	0.248	−2.001	0.2044	0.0493	0.01213	0.9967
	800	57.48	0.042	−1.609	0.0411	0.1085	0.02204	0.9955
	1000	57.86	0.989	−1.968	0.9148	0.1681	0.0426	0.9936
a*	200	−20.22	0.206	0.220	0.1139	0.0393	0.01691	0.9834
	400	−20.37	0.483	0.425	0.4224	0.1063	0.01007	0.9974
	600	−22.23	0.247	2.007	0.1050	0.0995	0.02409	0.9896
	800	−21.60	0.801	1.303	0.7495	0.1893	0.04443	0.9865
	1000	−23.15	1.424	3.014	1.0645	0.2100	0.04088	0.9945
b*	200	39.91	0.682	−0.201	0.6160	0.0491	0.02570	0.9649
	400	40.38	1.206	−0.644	0.7847	0.0516	0.01915	0.9865
	600	41.41	0.689	−1.651	0.5929	0.0603	0.01357	0.9960
	800	42.43	1.268	−2.699	0.7908	0.0683	0.01665	0.9972
	1000	42.05	1.028	−2.320	0.9718	0.1182	0.03778	0.9947
H_a_	200	116.97	1.286	−0.229	0.6394	0.0195	0.00263	0.9995
	400	116.89	0.973	−0.255	0.8587	0.1284	0.01605	0.9947
	600	118.55	1.394	−1.461	1.1385	0.1204	0.02589	0.9893
	800	118.03	1.163	−0.880	1.0976	0.2310	0.05563	0.9823
	1000	119.08	1.088	−2.171	0.7835	0.2635	0.04254	0.9946

Y_0_: the initial value of L*, a *, b * and H_a_, respectively; A: frequency factor; K(m^−1^): the rate coefficients of the color value change at a corresponding temperature; R^2^: correlation coefficient; σ: standard error.

**Table 2 foods-11-02540-t002:** Hunter L*, a* and b*, H_a_ and final temperature of green tea powder (GTP) with different thickness radiation with various times under microwave radiation under 700 W.

Sample Thickness (T_Th_, mm)	Radiation Time (T_I_, s)	Radiation Energy (W/g)	L*	a*	b*	Hue Angle *^a^* (H_ab_, ◦)	GTP Final Temp. (T_m_, °C)
20	180	0.34	49.87 ± 0.85	−7.26 ± 0.27	36.48 ± 0.44	100.65	132.8 ± 4.73
	150	0.28	50.71 ± 0.71	−10.91 ± 0.34	37.38 ± 0.49	105.70	113.6 ± 5.62
	120	0.22	51.61 ± 0.85	−13.45 ± 0.33	38.12 ± 0.61	110.10	95.3 ± 4.86
	90	0.17	52.58 ± 0.79	−16.42 ± 0.38	38.75 ± 0.54	112.71	78.2 ± 2.55
	60	0.11	53.61 ± 0.93	−18.45 ± 0.34	39.13 ± 0.57	114.64	62.7 ± 4.26
16	230	0.54	49.41 ± 0.71	−8.81 ± 0.24	36.77 ± 0.54	104.06	119.5 ± 4.16
	190	0.44	50.35 ± 0.67	−12.18 ± 0.27	37.59 ± 0.74	108.24	101.2 ± 3.08
	150	0.35	51.35 ± 0.80	−15.01 ± 0.26	38.27 ± 0.69	111.69	84.1 ± 2.03
	110	0.26	52.44 ± 0.92	−17.14 ± 0.48	38.81 ± 0.53	113.49	69.5 ± 2.59
	70	0.16	53.63 ± 0.98	−18.77 ± 0.54	39.17 ± 0.49	115.13	55.3 ± 3.10
12	280	0.87	49.69 ± 1.08	−13.49 ± 0.34	37.29 ± 0.57	109.70	96.7 ± 3.56
	230	0.72	50.29 ± 0.98	−15.75 ± 0.41	37.91 ± 0.65	112.91	80.5 ± 2.52
	180	0.56	51.37 ± 0.99	−16.83 ± 0.54	38.48 ± 0.51	113.80	75.6 ± 3.90
	130	0.41	52.46 ± 1.21	−18.17 ± 0.53	38.89 ± 0.79	114.89	62.8 ± 3.10
	80	0.25	53.68 ± 1.01	−19.01 ± 0.45	39.23 ± 0.56	115.85	52.7 ± 2.72
8	330	1.54	52.15 ± 1.01	−17.47 ± 0.38	37.44 ± 0.71	113.94	76.2 ± 3.00
	270	1.26	52.47 ± 1.04	−18.51 ± 0.45	38.46 ± 0.58	115.36	65.4 ± 2.95
	210	0.98	53.06 ± 0.83	−19.07 ± 0.28	39.08 ± 0.62	115.94	56.2 ± 3.44
	150	0.70	53.79 ± 0.83	−19.47 ± 0.31	39.3 ± 0.65	116.07	49.2 ± 2.78
	90	0.42	54.59 ± 0.71	−19.71 ± 0.38	39.49 ± 0.68	116.16	45.7 ± 2.29
4	380	3.55	53.44 ± 0.84	−18.61 ± 0.42	38.81 ± 0.61	115.09	53.7 ± 3.95
	310	2.90	53.51 ± 0.82	−19.32 ± 0.38	39.24 ± 0.71	116.19	45.3 ± 2.25
	240	2.24	53.94 ± 0.76	−19.68 ± 0.49	39.44 ± 0.80	116.66	39.7 ± 1.84
	170	1.59	54.46 ± 0.91	−19.86 ± 0.31	39.59 ± 0.45	116.66	35.2 ± 1.73
	100	0.93	55.01 ± 0.91	−19.95 ± 0.38	39.68 ± 0.60	116.70	31.5 ± 1.37
CK	0	0.00	55.85 ± 0.80	−20 ± 0.45	39.7 ± 0.72	116.74	

*^a^* calculated with the average values of a* and b*.

**Table 3 foods-11-02540-t003:** The apparent activation energy required for color change in different thicknesses of green tea powder.

Color Parameters	E_a_ (KJ/mol) at Different H_Th_ (mm)					Fitting	
	20	16	12	8	4	Model	R^2^
a*	12.15	16.32	25.95	38.23	56.24	E_a_ = −0.044 + 83.6 exp(−0.011 H_T__h_)	0.9989
b*	12.67	16.32	32.98	53.98	99.63	E_a_ = 4.32 + 179.7 exp(−0.159 H_T__h_)	0.9981
H_a_	0.84	6.18	22.51	56.22	65.59	E_a_ = −80.4 + 164.1 exp(−0.041 H_T__h_)	0.9514
L*	6.59	10.09	29.55	47.28	82.67	E_a_ = −8.43 +146.3 exp(−0.119 H_T__h_)	0.9943

**Table 4 foods-11-02540-t004:** The contents of the catechins (EC, EGC, ECG, and EGCG), total free amino acids (FAA), reduction sugars (RS), and theaflavin (TF) in the green tea powder samples treated by microwave radiation of 1000 W with various time.

Time (min)	EC (%)	EGC (%)	ECG (%)	EGCG (%)	FAA (%)	RS (%)	TF (%)
0	0.68 ± 0.008 ^a^	2.86 ± 0.041 ^a^	0.89 ±0.016 ^a^	5.24 ± 0.074 ^a^	5.01 ± 0.052 ^a^	1.04 ± 0.021 ^a^	0.00
4	0.67 ± 0.012 ^a^	2.86 ± 0.052 ^a^	0.89 ±0.018 ^a^	5.24 ± 0.091 ^a^	4.91 ± 0.067 ^a^	1.03 ± 0.024 ^a^	0.00
6	0.67 ± 0.018 ^a^	2.86 ± 0.061 ^a^	0.89 ±0.021 ^a^	5.24 ± 0.084 ^a^	4.81 ± 0.071 ^a^	1.04 ± 0.016 ^a^	0.00
9	0.68 ± 0.017 ^a^	2.86 ± 0.063 ^a^	0.89 ±0.022 ^a^	5.24 ± 0.098 ^a^	4.62 ± 0.063 ^ab^	1.04 ± 0.019 ^a^	0.00
12	0.60 ± 0.014 ^b^	2.56 ± 0.057 ^b^	0.91± 0.018 ^a^	5.02 ± 0.110 ^a^	4.44 ± 0.056 ^b^	1.02 ± 0.018 ^a^	0.011 ± 0.008 ^a^
15	0.37 ± 0.008 ^c^	1.79 ± 0.036 ^c^	0.75 ±0.017 ^b^	3.74 ± 0.075 ^b^	4.04 ± 0.052 ^c^	0.88 ± 0.017 ^b^	0.008 ± 0.003 ^b^

Data are presented as mean ± standard error of mean (SEM, n = 3). Mean values with the different lowercase letters in the same column indicate a significant difference with the least significant difference (LSD) test (*p* < 0.05).

## Data Availability

Data is contained within the article or Appendix A.

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
