# Peer review of "The Effect of Microwave Radiation on the Green Color Loss of Green Tea Powder"

_foods, 2022, doi:10.3390/foods11162540_

Round 1
Reviewer 1 Report
See comments on the MS file

Author Response
- Before the results you should describe your experimental design. What was your experimental design, technical and biological replication?
Re: We have added the information in the abstract.
- The abstract is missing experimental design, objectives and conclusion needs to be rephrased so as to reflect original objectives.
Re: We have added the information in the abstract to reflect original objectives.
- Should be alphabetically
Re: Revised.
- The introduction lacks any review of existing literature on selection indices. This cannot be ignored and it should be included.
Re: We have added the information to describe the situation.
5. What was your experimental design?
Author must describe the experiments with much more details because a reader has to understand absolutely clear your experimental design and has to be able to repeat it. It is unclear and should be written clearly
Re: The details of the experimental information has been added in section 2.2.
- How many levels or times of microwave radiation?
Re: The information is added in section 2.3.
- How many levels or times of microwave radiation?
Re: The information is added in section 2.4.
- Ref?
Re: We first report the measured value could not meet the real color, and calibrate the values using the method for the first time. So we cann’t provide Ref.
- which or How many temperatures?
Re: The temperature values were showed in Table 2. Each group has 5 temperatures.
- year
Re: Added.
- What was your standard sample, concentrations and standard curve?? & 12. How about your standard sample and concentration and standard curve?
Re: The standard samples are given in the section 2.1. Here we do not present detail information of concentrations and standard curve since this article does not aim to determination method.
- this part is result and discussion section and you should describe your results not material and method. should be removed.
Re: Revised accordingly.
- First you should describe your ANOVA results and then mean comparison of groups.
Re: Revised accordingly.
- you should discuss about your results. discussion is not report.
Re: Some sentances are added to discuss the results.
- How about the replication and error bar (SEM) and mean comparison of radiation times?
Re: Three replications of each experiment were performed, and the reproducibility of the assayed values was within the range of 2.5%. We give this information in section 2.2. The mean comparison did not perform since we aim to study the kinetics of the color loss.
17. The table should be improved. it is not completed information.
Re: We supplment the data in Table 2.
18. the title should be summarised such as effect of different activation energy on green tea powder
Re: Revised accordingly.
19. you should describe your own results
Re: Revised accordingly.
20. First you should describe your ANOVA results and then mean comparison of groups. & 21. First you should describe your ANOVA results and then mean comparison of groups.
Re: Revised accordingly.
22. Please show mean comparison results in each groups with alphabets
Re: Added accordingly in Fig. 3.
23. should be changed to SEM. standard deviation for mean comparison of groups is wrong.
Re: Revised accordingly.
Reviewer 2 Report
The work done by authors is good. how ever following changes needed.
1- compare the role of MW with other rays
2- if possible give FTIR before and after treatment of extract
3- improve literature survey add recent work
Author Response
- compare the role of MW with other rays
Re: That could be interesting if comparing the role of MW with other rays such as ultrasonic and infrared wave. However, that is not the research content belonging to our topic.
- if possible give FTIR before and after treatment of extract
Re: At the moment, we cann’t performaced the FTIR of the samples before and after treatment of extract.
- improve literature survey add recent work
Re: Already added literature from searchs with google scholar.
Reviewer 3 Report
Methodology section
It would be important to specify or clarify, why did you work with two microwave equipment? What was the objective?, only to obtain a wide range of powers applied on the sample?
Paragraph 71, clarify or specify how the times for the treatments at 700W were determined, in the same way for the equipment in witch different powers were tested.
It would be important to make clear from the methodology all the treatments that were used on in the research. The term CK appears up to the results section, referring to Figs. 1A and 1B but is not described in M&M section.
In paragraph 170, there seems to be an mistake in the nomenclature in fig. 1 relative to b*
Results
In table 2. Please indicate in the caption that data in this table were obtained using 700W.
Table 2. There were no discussion of the results obtained for CK.
In the line 324 and 325 “These results indicated that the green color loss in GTP during food processing and microwave pasteurization is closely correlated with heating time and sample thickness”. To reach this conclusion, it would be necessary to carry out the analyses in the food as is traditional consumed (infused tea).
Author Response
1. It would be important to specify or clarify, why did you work with two microwave equipment? What was the objective?, only to obtain a wide range of powers applied on the sample?
Re: Thank you very much for your question. For the study of the effect of output power on green color lose, we tried to obtain a wide range of powers (200 -1000 W) applied on the samples. As we know the output power of the microwave oven are adjusted by run-time. When run the highest power (100%) the microwave are output in a continuous manner, which is convinient for us to control the run time in the study of the effect of microwave radition on the samples with different thickness. Our pre-experiments showed 1000 W was too high for the study. So we applied another microwave oven with 700 W.
2. Paragraph 71, clarify or specify how the times for the treatments at 700W were determined, in the same way for the equipment in witch different powers were tested.
Re: We are sorry for the un-clarified description. The sentences are rewrriten.
3. It would be important to make clear from the methodology all the treatments that were used on in the research. The term CK appears up to the results section, referring to Figs. 1A and 1B but is not described in M&M section.
Re: The information of CK are added in M&M section.
4. In paragraph 170, there seems to be an mistake in the nomenclature in fig. 1 relative to b*
R: Thanks a lot. We correct the the nomenclatures in Fig. 1.
Results
5. In table 2. Please indicate in the caption that data in this table were obtained using 700W.
R: The infromation is added now.
6. Table 2. There were no discussion of the results obtained for CK.
R: Two sentences are added to discuss the result.
7. In the line 324 and 325 “These results indicated that the green color loss in GTP during food processing and microwave pasteurization is closely correlated with heating time and sample thickness”. To reach this conclusion, it would be necessary to carry out the analyses in the food as is traditional consumed (infused tea).
Re: We revised the description.